

# Design of information management model based on multiobjective optimization algorithm in intelligent electric financial system

Junhui Hu[1], Hongxiang Cai[2], Shiyong Zhang[2], Chuanxun Pei[2] and Zihao Wang[2]

[1] State Grid Ningbo Electric Power Supply Company, Ningbo, Zhejiang, China
[2] State Grid Ninghai Power Supply Company, Ningbo, Zhejiang, China

Corresponding author
Junhui Hu, hujunhui1122@163.com

## ABSTRACT

The electric power infrastructure is the cornerstone of contemporary society's sustenance and advancement. Within the intelligent electric power financial system, substantial inefficiency and waste in information management persist, leading to an escalating depletion of resources. Addressing diverse objectives encompassing economic, environmental, and societal concerns within the power system helps the study to undertake a comprehensive, integrated optimal design and operational scheduling based on a multiobjective optimization algorithm. This article centers on optimizing the power financial system by considering fuel cost, active network loss, and voltage quality as primary objectives. A mathematical model encapsulates these objectives, integrating equations and inequality constraints and subsequently introducing enhancements to the differential evolutionary algorithm. Adaptive variation and dynamic crossover factors within crossover, variation, and selection operations are integrated to optimize algorithm parameters, specifically catering to the multiobjective optimization of the electric power system. An adaptive grid method and cyclic crowding degree ensure population diversity and control the Pareto front distribution. They experimentally validated the approach and the comparisons conducted against AG-MOPSO, INSGA-II, and NSDE algorithms across standard test functions: ZDT1, ZDT2, ZDT3, and DTLZ4. The convergence evaluation indices for this study's scheme on ZDT1 and ZDT2 are 0.000938 and 0.0034, respectively. Additionally, distribution evaluation indices on ZDT1, ZDT2, ZDT3, and ZDT4 stand at 0.0018, 0.0026, 0.0027, and 0.0009, respectively. These indices indicate a robust convergence and distribution, facilitating the optimization of electric power financial information management and the intelligent handling of the electric power financial system's information, thereby enhancing the allocation of material and financial resources.

# INTRODUCTION

The swift evolution of the electric power industry, coupled with heightened levels of information integration, underscores the pivotal significance of optimizing and managing information within the electric power financial system. The informatization of financial management in this sector facilitates multidimensional data analysis and decision-making support by seamlessly integrating and sharing data with other business systems. This information ensures data uniformity and integrity, amplifying management efficacy and decision-making acumen. However, the surge in data volume due to the information explosion necessitates more robust management within the electric power financial information system.

To cater to the demands of corporate decision-making, the financial management information technology within the power industry necessitates automation and intelligence. This technology should encompass automated data acquisition, processing, and analysis capabilities, as well as intelligent decision-making and early warning systems based on preset rules and models. This strategic shift aims to curtail manual intervention and errors, augmenting management efficiency and accuracy. The multiobjective optimization algorithm (*Abdollahzadeh & Gharehchopogh, 2022*) emerges as a potent optimization technique adept at handling multiple conflicting objective functions. Given the intricate nature of the electric power financial system, replete with interwoven objectives and factors spanning power cost, reliability, security, and customer service quality, employing multiobjective optimization algorithms becomes pivotal. These algorithms resolve intricate power financial challenges by optimizing numerous objective functions, deriving optimal solutions, and tackling complex problems (*Liu & Wang, 2019*).

Moreover, the intelligent power financial system serves as an internal information management system within electric power enterprises, primarily catering to financial management and decision support. This system amalgamates data and information from the power scheduling and information management system, tailoring its analysis and processing capabilities to suit financial management and decision-making requisites. Through the synthesis and analysis of power scheduling information management data, the intelligent power financial system aids power enterprises in conducting power costing, energy market analysis, and business decision-making, ultimately enhancing enterprise management proficiency and decision-making efficiency (*Feng, Shi & Wang, 2021*).

Within the intricate landscape of power energy scheduling, numerous complex multiobjective optimization problems persist, with large-scale power system scheduling emblematic of these challenges. Inefficient power dispatching will lead to the delayed generation of financial reports and affdecision makers' timely understanding of the company's financial status and operating resources. In emergencies, such delays can lead to missed opportunities or an inability to respond to risks promptly. At the same time, it may also lead to wrong financial decisions, such as bad investment decisions, improper budget allocation, *etc*. These bad decisions can lead to significant economic losses and even affect the company's long-term development. Consequently, considering cost, resources, and other facets inherent in power scheduling within the financial system, we devise an

information management model using a multiobjective optimization algorithm for the intelligent power economic system, aiming to address the multiobjective optimization quandaries within the power system. This article's principal contributions are delineated as follows:

1. Establishment of an information management model: This study employs the fuel cost of thermal power generating units, the active network loss of the power system, and voltage quality as the pivotal objectives for optimization within the intelligent power financial system. Mathematical models for each purpose are formulated, incorporating equation and inequality constraints. Moreover, state variables in the power system optimization are restricted using penalty functions.

2. Enhancement of the multiobjective differential evolutionary algorithm: Building upon the principles of the differential evolution algorithm, this article introduces the multiobjective differential evolution algorithm. This innovative algorithm addresses the multiobjective optimization of the power system by integrating adaptive mutation and dynamic crossover factors within crossover, mutation, and selection operations.

3. Construction of a process framework for multiobjective optimization of the power financial system: Employing an adaptive grid method alongside cyclic crowding degree ensures population diversity and governs the distribution of the Pareto front. These methods are rigorously tested against standard test functions, validating the efficacy and performance of the proposed process framework.

## RELATED WORKS

### Multiobjective optimization

Initially, research scholars introduced heuristic algorithms, also known as intelligent optimization algorithms (*Yang et al., 2020*). These algorithms are rooted in an iterative stochastic optimization framework, leveraging natural and life phenomena to formulate algorithmic logic to solve single or multiple objective optimizations. Early heuristic algorithms encompassed genetic algorithms (*Sohail & Ayesha, 2023*), particle swarm optimization algorithms (*Gad & Ahmed, 2022*), differential evolutionary algorithms (*Wang et al., 2022*), and, more recently, pedagogical optimization algorithms (*Yu et al., 2021*). Primarily designed for solving single-objective problems, these algorithms excel in achieving optimization outcomes centered around a singular objective. Subsequently, scholars advocated for multiobjective optimization theory, stemming from its origins in economic theory. This concept originated with the Italian-American economist Pareto, who introduced the concept of Pareto optimality (*Bae et al., 2023*). Over time, this theory evolved into a comprehensive technique—multiobjective optimization problems (MOP). MOPs address scenarios involving more than one, often conflicting, optimization objectives.

Among the Pareto optimization-based methods, a prominent set of algorithms in engineering applications includes the family of genetic algorithms that utilize non-dominated sorting, as proposed in *Fu & Liu (2019)*. This approach incorporates non-dominated sorting to organize populations and integrates it with a genetic algorithm

based on Goldberg's principles. Additionally, the literature introduces a microhabitat and morphological approach for optimal individual selection aimed at identifying multiple Pareto optima. While this method offers the advantage of uniformly distributed optimal solutions, it suffers from decreased computational efficiency and heightened complexity and needs an elite mechanism. Subsequently, *Wang et al. (2019)* addressed the deficiencies of the NSGA algorithm (*Lu et al., 2019*) and introduced the NSGA-II algorithm. NSGA-II enhances the original algorithm by introducing a crowding comparison operator, evaluating the crowding degree among individuals within the same non-dominated layer, and selecting individuals with a higher crowding degree. Furthermore, NSGA-II incorporates an elite strategy, broadening the sampling space and significantly improving the algorithm's solution-finding capabilities and speed while preserving population diversity and reducing algorithmic complexity.

Despite NSGA-II's significant advancements, its applicability is broader than scenarios involving three or more objectives. To tackle this, *Miriam, Swaminathan & Chakaravarthi (2021)* introduced the NSGA-III algorithm, employing a reference point-based approach within the NSGA-II framework. NSGA-III utilizes association and small habitat techniques in individual selection, effectively optimizing scenarios involving 3 to 15 objectives, yielding superior Pareto solutions. These algorithms stand out for their substantial advantage in addressing real engineering multiobjective optimization problems and have garnered recognition within the industry.

Another category of algorithms, distinct from Pareto optimization-based approaches, involves the decomposition-based multiobjective optimization method, as presented in the *Peng & Ishibuchi (2020)*. This method adopts a scalarization concept, breaking down the multiobjective problem into several scalar subproblems. It concurrently optimizes each subproblem while considering its neighboring subproblems, eventually deriving the optimal solution. Renowned for its efficiency and accuracy, this method is one of the most notable heuristic multiobjective algorithms.

Compared to the NSGA-II algorithm, this approach showcases time complexity and convergence advantages, particularly in scenarios involving low-dimensional objectives. Additionally, it exhibits superior uniformity. However, in some high-dimensional cases, its uniformity slightly lags behind NSGA-II.

## Optimization of the electric power system

Optimization challenges within power systems have commanded scholarly attention since the late 20th century. Over subsequent decades, power system optimization has been a focal point in academic research and industry endeavors. This intricate problem encompasses multiple objectives, various time scales, and diverse operational states (*Xu et al., 2020*). Power system scheduling tasks are broadly categorized into four types based on distinct optimization levels and time scales: economic load dispatch (ELD) (*Singh et al., 2023*), unit commitment (UC) (*Egbue et al., 2022*), optimal power flow (OPF) (*Risi, Riganti-Fulginei & Laudani, 2022*), and distributed power generation (DPG) (*Ismael et al., 2019*) encompassing siting and capacity settings (*Ismael et al., 2019*).

For economic load dispatch (ELD), *Premkumar et al. (2020)* introduced a multiobjective pedagogical optimization algorithm, incorporating a non-dominated sequencing method. This approach effectively addresses optimal economic scheduling and minimum emission scheduling problems. Additionally, enhancing existing multiobjective PSO algorithms has proven beneficial. *Zhou, Wang & Chai (2022)* presents the MOPSO algorithm combined with fuzzy adaptive techniques and self-learning strategies. In contrast, *Kong, Wang & Zhao (2021)* proposes the MOPSO algorithm with objective weight orientation, adept at determining optimal scheduling solutions within practical constraints.

Concerning unit commitment (UC), *Zakaria et al. (2020)* utilizes a stochastic modeling approach for the unit combination problem, offering a corresponding optimization tool for optimal solutions. Another work, *Akay et al. (2021)*, integrates the modal evolutionary algorithm and NSGA-II algorithm to manage the unit operating state, and this involves a local search strategy and applies weighted and λ-iteration methods for optimization. Moreover, *Zouache et al. (2023)* enhances the artificial bee colony algorithm, applying fuzzy binary actual number coding to optimize system reliability, fuel cost, and emission (*Kaya et al., 2021*).

In the optimal power flow (OPF) domain, *Ali et al. (2023)* enhances the PSO algorithm by introducing the chaotic queueing method and adaptive concepts to adjust IPSO parameters, and it facilitates optimal voltage stability solutions with objectives centered around cost and emission. Another contribution, *Alomoush et al. (2022)*, introduces the dyadic modified Jaya method (QOJaya), employing intelligent dyadic learning, fuzzy strategies, and an external elite pool preservation method for trend optimization. Further, *Afshari, Hare & Tesfamariam (2019)* improves the artificial bee colony algorithm, proposing the fuzzy modified artificial bee colony (MABC) for enhanced performance. For Distributed Generation Siting and Capacity Setting, *Chen, Fang & Zhong (2022)* integrates the NSGAH I algorithm with the point estimation method (PEM) to achieve optimized outcomes considering total cost, total network loss, and customer outage costs as optimization objectives, employing an integrated multiobjective optimization method.

Conventional approaches often convert complex problems into single-objective optimizations through techniques like weighting or linear programming in the realm of multiobjective optimization in power systems. However, this approach poses challenges in weight selection and fails to effectively address the multiple conflicting objectives inherent in practical engineering scenarios. Thus, this article embarks on a novel trajectory, aiming to enhance the efficacy of multiobjective optimization algorithms by leveraging differential evolutionary algorithms.

## METHODOLOGY

This article focuses on three critical objective functions: fuel cost, active network loss in the electric power system, and the voltage quality of thermal power generating units, as depicted in Fig. 1. We constrain the state variables in the power system optimization by imposing equation and inequality constraints on these three objectives. We enhance the multiobjective differential evolutionary algorithm by utilizing this

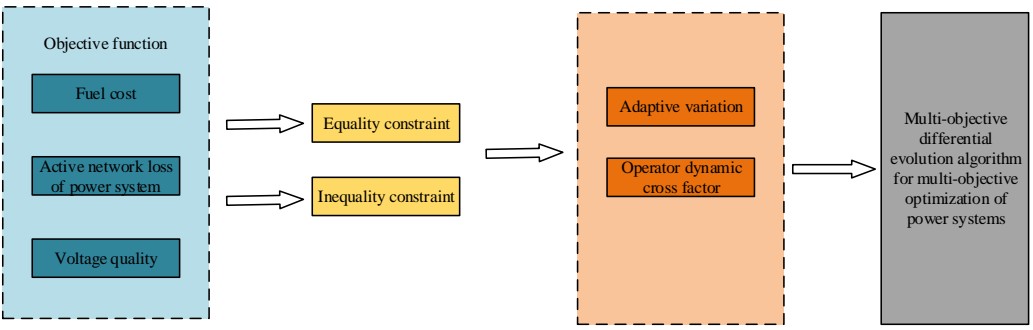

**Figure 1** Overall model structure.

information management mathematical model as a foundation. This enhancement involves leveraging the principles of the differential evolutionary algorithm, incorporating adaptive variation and dynamic cross-factor operators. Consequently, we construct a multiobjective differential evolutionary algorithm tailored explicitly for solving power system multiobjective optimization.

Furthermore, we employ the adaptive grid method and cyclic congestion degree to ensure population diversity and govern the distribution of the Pareto frontier. This endeavor aims to realize the electric power financial system's information management model design based on the multiobjective differential evolutionary algorithm.

## Mathematical model of information management

This article's information management mathematical model comprises three primary components: an objective function, equation constraints, and inequality constraints. The fundamental form of the mathematical model is articulated as follows:

$$\begin{cases} \min f(u,x) \\ g(x) = 0 \\ h(u,x) \leq 0 \end{cases} \tag{1}$$

Here, 'f' signifies the objective function, 'g' denotes the equality constraint, and 'h' represents the inequality constraint. 'u' pertains to control variables, while 'x' corresponds to state variables. We embark on the mathematical modeling of the objective function for the current fuel cost, active network loss magnitude, and voltage quality data within the electric power financial system. The fuel cost is usually proportional to the generator's fuel consumption. The fuel cost function may differ for each generator type (coal, gas, nuclear, *etc*). Suppose we have multiple types of generators, each with its specific fuel cost function. Active network loss is the energy loss due to the network's current flow. These losses are usually proportional to the resistance of the square sum of the current. Voltage quality is an essential parameter in a power system, which directly impacts the safe operation of equipment and the user's power experience. Voltage quality is usually evaluated by voltage deviation, fluctuation, and flicker. Initially, we adopt the fuel cost of thermal generating

units as the objective function, employing a quadratic function in the following manner:

$$\min f_1 = \min \sum_{i=1}^{N_G} a_i + b_i P_i + c_i P_i^2 \tag{2}$$

where $a_i, b_i$ and $c_i$ are the cost coefficients expressed as a quadratic function of unit i. $N_G$ is the number of generators, and $P_i$ is the active output of unit i, including the generators at the balancing node. Simultaneously, owing to the unavoidable loss of power during the transmission of electrical energy—wherein the active power consumed by resistors and conductors dissipates as heat energy—optimization of this active loss becomes imperative within the information management system. By computing and minimizing the active loss, optimization in allocation can be achieved, thereby enhancing operational efficiency and bolstering the profitability of power finance. Consequently, the second objective function chosen revolves around the active power loss, articulated as follows:

$$\min f_2 = \min \sum_{k=1}^{m} G_{k(i,j)}(U_i^2 + U_j^2 + 2U_i U_j \cos\theta_{ij}) \tag{3}$$

where m is the number of branches of the system, $G_{k(i,j)}$ is the loss of active power generated in the electric line from node k node i to node j of the branch. In this article, the voltage is expressed in polar coordinates. $U_i, U_j$ are the voltage of node i and j, respectively, $\theta_{ij}$ and are the phase angle difference of the voltage of node i and j. Within the information management framework of the electric power financial system, it becomes imperative to document the voltage value at each node during the operation of the power system. Excellent voltage stability at intermediate positions signifies enhanced voltage quality, minimizing impacts on equipment and end-users. Hence, we establish an additional objective function addressing this criterion:

$$\min f_3 = \min \sum_{k=1}^{n} \left( \frac{2U_i - U_{i,\max} - U_{i,\min}}{U_{i,\max} - U_{i,\min}} \right)^2 \tag{4}$$

where n is the number of nodes in the system, $U_{i,\max} U_{i,\min}$ the maximum and minimum voltage values allowed at node i are, respectively, and $U_i$ the actual voltage is at the node.

Then, we formulated constraints within the mathematical model to optimize information management in the power financial system. These constraints manifest in two categories: equational and unequal constraints. Equational constraints primarily involve active and reactive power balance, while unequal constraints encompass node voltage, generator active and reactive power, reactive power compensation, and transformer tapping position constraints. We articulate the equation constraints and inequality constraints as delineated below:

$$\begin{cases} P_{Gi} - P_{Li} = U_i \sum_{j=1}^{n} U_j (G_{ij} \cos\theta_{ij} + B_{ij} \sin\theta_{ij}) \\ Q_{Gi} - Q_{Li} = U_i \sum_{j=1}^{n} U_j (G_{ij} \cos\theta_{ij} - B_{ij} \sin\theta_{ij}) \end{cases} \tag{5}$$

$$\begin{cases} U_{i,\min} \le U_i \le U_{i,\max}, i \in N \\ P_{Gi,\min} \le P_{Gi} \le P_{Gi,\max}, i \in N_G \\ Q_{Gi,\min} \le Q_{Gi} \le Q_{Gi,\max}, i \in N_G \\ T_{i,\min} \le T_i \le T_{i,\max}, i \in N_T \\ Q_{ci,\min} \le Q_{ci} \le Q_{ci,\max}, i \in N_c \end{cases} \quad (6)$$

In Eq. (5), the equation constraints are defined, where $P_{Gi}$ and $Q_{Gi}$ represent the active and reactive outputs of node i. $P_{Li}$ and $Q_{Li}$ represent the active and reactive demands of node i, respectively. n is the set of nodes connected to node i. The inequality constraints are defined in Eq. (6), where $U_{i,\max}$ and $U_{i,\min}$ are the maximum and minimum voltage values at node i, respectively. N is the set of nodes in the system. $P_{Gi,\max}P_{Gi,\min}$. They are the maximum and minimum values of the active output of generation node i. $Q_{Ci,\max}$ and $Q_{ci,\max}$ are the upper and lower limits of reactive power compensation node i, respectively. NC is the collection of reactive power compensation nodes.

We construct penalty functions for state variable constraints to further intelligently constrain the state variables. Let $\lambda_P, \lambda_U$ and $\lambda_Q$ be the penalty factors, P represents the active output of the balancing node, $U_i$ denotes the voltage of the load node i, $N_Q$ isthe set of load nodes, $Q_j$ isthe reactive output of the generating node j, and $N_G$ is the set of generating nodes, at this point the penalty function PF is constructed as follows:

$$PF = \lambda_P \left( \frac{P - P_{vl}}{P_{\max} - P_{\min}} \right) + \lambda_U \left( \frac{U_i - U_{i,vl}}{U_{i,\max} - U_{i,\min}} \right)^2 + \lambda_Q \left( \frac{Q_j - Q_{j,vl}}{Q_{j,\max} - Q_{j,\min}} \right)^2 \quad (7)$$

Eventually, we construct the information management model after adding the penalty function PF to the three objective functions mentioned above.

## Improved multiobjective differential evolutionary algorithm

Solution through the iterative evolution of intelligent algorithms. In this section, we adopt the multiobjective differential evolutionary algorithm for evolution. However, prevailing algorithms exhibit limitations, rendering them unsuitable for designing information management models in electric power financial systems. Hence, we propose enhancing the differential evolutionary algorithm rooted in the dominance relationship to address multiobjective problems within the electric power financial system.

Recognizing the structural resemblance between the general differential evolutionary algorithm and genetic algorithms, both populations aim to approach the problem's optimal solution through crossover, mutation, and selection operations. Divergence arises in their strategies for crossover and mutation. At the same time, the selection operation in this study employs the INSGA-II algorithm (*Chen, Fang & Zhong, 2022*) for stratifying the population of individuals based on dominance relationships and cyclic congestion calculations. This stratification facilitates the operation of superior individuals within the population, as depicted in Fig. 2. The process involves retaining dominant individuals for the subsequent generation, ultimately acquiring the Pareto optimal solution set after continuous evolution.

Based on the scope and characteristics of the target space, a set of grids is initialized, and each solution in the population is assigned to the corresponding grid. The mesh size

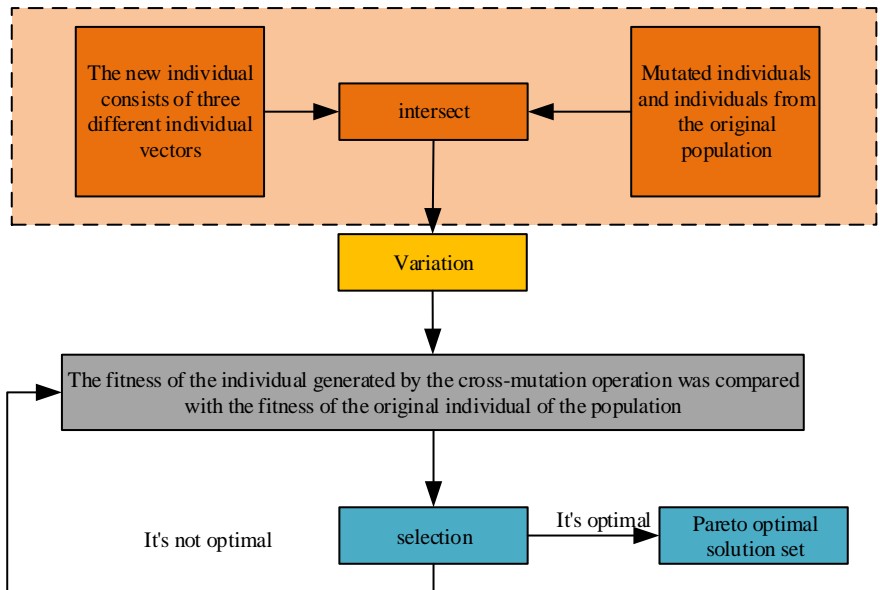

**Figure 2** Crossover, variation and selection process of differential evolution algorithm.

is dynamically adjusted according to the distribution and density of the solutions. If the solutions in a grid are too dense, you can reduce the grid size to increase the search detail. If there are no or few solutions in a grid, you can increase the grid size to expand the search scope. In a selection operation, such as selecting the next-generation population, the solution located in the sparse grid is preferentially chosen to preserve the diversity of the population. Each non-dominated solution's degree of crowding in the target space is calculated. The degree of crowding can be calculated by considering the difference in the target value of a solution between its neighbors. The non-dominated solutions are then sorted according to the degree of crowding. When selecting the next-generation population, the solution with a greater crowding degree is preferentially chosen to ensure the uniform distribution of solutions on the Pareto front. In the process of algorithm iteration, the congestion value is constantly updated to reflect the distribution of solutions, and this can be done by recalculating the congestion level after each iteration.

We set the formula for the crossover as shown below:

$$V_i(t+1) = X_{r1}(t) + F \cdot (X_{r2}(t) - X_{r3}(t)) \tag{8}$$

wherein V represents the vector of newly generated individuals, X denotes the vector containing individuals within the population, i signifies the i-th generated individual, t denotes the t-th iteration, and 'F' represents the variation factor. The variables 'r1', 'r2', and 'r3' are distinct positive integers selected randomly to denote three different vectors of individuals chosen from the population. One of these vectors serves as the base vector. In contrast, the difference vectors of the other two individuals are weighted and added to the base vector to derive the mutated individuals' vector. The randomness in selecting individuals from the parent population contributes to varied combinations, augmenting

the population's diversity. Notably, during the initial iterations, the disparity among individuals is substantial, thereby bolstering robust global search capabilities. However, as iterations progress, the discrepancy between individuals diminishes, enhancing the algorithm's aptitude for local convergence.

To further augment the population's diversity, individuals slated for crossover predominantly comprise the mutated individuals described earlier and those within the original population. The crossover formula is articulated as follows:

$$u_{ij}(t+1) = \begin{cases} v_{ij}(t+1), rand(j) \leq CR\, or\, j = h \\ x_{ij}(t), rand(j) > CR\, or\, j \neq h \end{cases} \tag{9}$$

where 'u' denotes the individual after the crossover operation, 'v' denotes the individual after the mutation operation, and 'x' denotes the individual within the parent population. CR represents the crossover probability factor, 't' represents the t-th iteration, 'i' signifies the ith individual, where the total number of individuals equals the population size; 'j' denotes the j-th dimensional component, and 'h' is chosen as an integer ranging from 1 to the number of variable dimensions and this ensures that the crossover-operated individual retains at least one-dimensional component from the mutated individual.

The crossover probability factor determines the selection of components: if the random number is less than or equal to CR, the element from the variant individual is utilized; otherwise, the component from the original individual is retained for the operation.

Finally, a selection operation akin to genetic algorithms ensues, wherein the fitness of individuals produced by the crossover-mutation operation is compared to the fitness of the original individuals within the population. Superior individuals are selected to progress into the subsequent generation. The algorithm iterates until it reaches the upper limit of iteration, leading the population toward or approximating the optimal solution. Throughout these operations—crossover, mutation, and selection—we introduce an adaptive mutation operator within the differential evolution algorithm. This adaptation aims to prevent the algorithm from converging into a local optimum owing to a reduction in population diversity:

$$\begin{cases} \lambda = \exp(1 - \dfrac{T}{T+1-t}) \\ F = F_0 \times 2^{\lambda} \end{cases} . \tag{10}$$

In the formula, T denotes the total number of iterations, and t represents the current number. F denotes the variation operator; when the algorithm is initialized with t being 0, F is close to $2F_0$, and as the iterations proceed, F decreases gradually and finally approaches $F_0$. Secondly, when its value is significant, the proportion of variation in the crossed individuals is larger, which is suitable for local search, and relatively, when its value is small, the proportion of the original population is larger. The diversity of the population is better for global search. Therefore, the dynamic crossover factors we took are as follows:

$$CR = CR_{\min} - \frac{t \cdot (CR_{\max} - CR_{\min})}{T}. \tag{11}$$

The formula $CR_{\max} CR_{\min}$ contains the crossover factor's maximum and minimum values. Upon initialization of the algorithm, the cross-factor CR begins at its minimum

value. Throughout subsequent iterations, CR progressively approaches its maximum value. This progressive adjustment prioritizes global search in the algorithm's initial stages while focusing on local search in its later stages.

## Electric power financial system

Next, we combine 'Mathematical model of information management' and 'Improved multiobjective differential evolutionary algorithm' to carry out multiobjective optimization of the electric power financial system. The specific process is shown in Fig. 3, and its operations are as follows:

Step 1: Define algorithm parameters, initialize each control variable, and set the iteration count to $i = 0$.

Step 2: Conduct calculations based on variable values. Utilize the calculated state variable values to derive the sub-goal function values for each individual. By assessing sub-objective fitness, the selection operation is executed by employing dominance relationship calculation and crowding. Retain NP individuals for the subsequent generation.

Step 3: Evaluate if the termination condition is met. If affirmative, proceed to step (5); otherwise, continue with the subsequent operation.

Step 4: Determine the number of non-dominated solutions (level) value and compare it against the population size. Based on this comparison, implement the three mutation strategies mentioned in distinct scenarios. Subsequently, crossover operations will be performed to enhance population diversity. Following mutation and crossover, integrate the resultant population with the original one and return to step (2). Increment the iteration count: $i = i + 1$.

Step 5: Conclude the algorithm's iterations. Retain nRep solutions designated as the Pareto optimal solution set, determined through cyclic congestion calculations.

## EXPERIMENTS AND ANALYSIS

This section uses Electric Power Assisted Steering data collected from the HIL at Volvo (https://zenodo.org/records/3263796). We conduct experimental comparisons and analyses involving AG-MOPSO (*Kong, Wang & Zhao, 2021*), INSGA-II (*Chen, Fang & Zhong, 2022*), and NSDE (*Fu & Liu, 2019*) algorithms. We select four test functions to gauge algorithm performance—ZDT1, ZDT2, ZDT3, and DTLZ4—each featuring distinct Pareto frontiers. The decision variable space comprises 30 dimensions for the first three functions and 12 dimensions for the latter. These functions serve as benchmarks to evaluate and compare the algorithms' performance.

### Evaluation indicators

We selected model evaluation indices categorized into convergence and distribution metrics. Generational distance (GD) is the selected convergence evaluation index. The specific expression for GD is represented by Eq. (12):

$$GD = \frac{1}{n}\sqrt{\sum_{i=1}^{n} d_i^2} \tag{12}$$

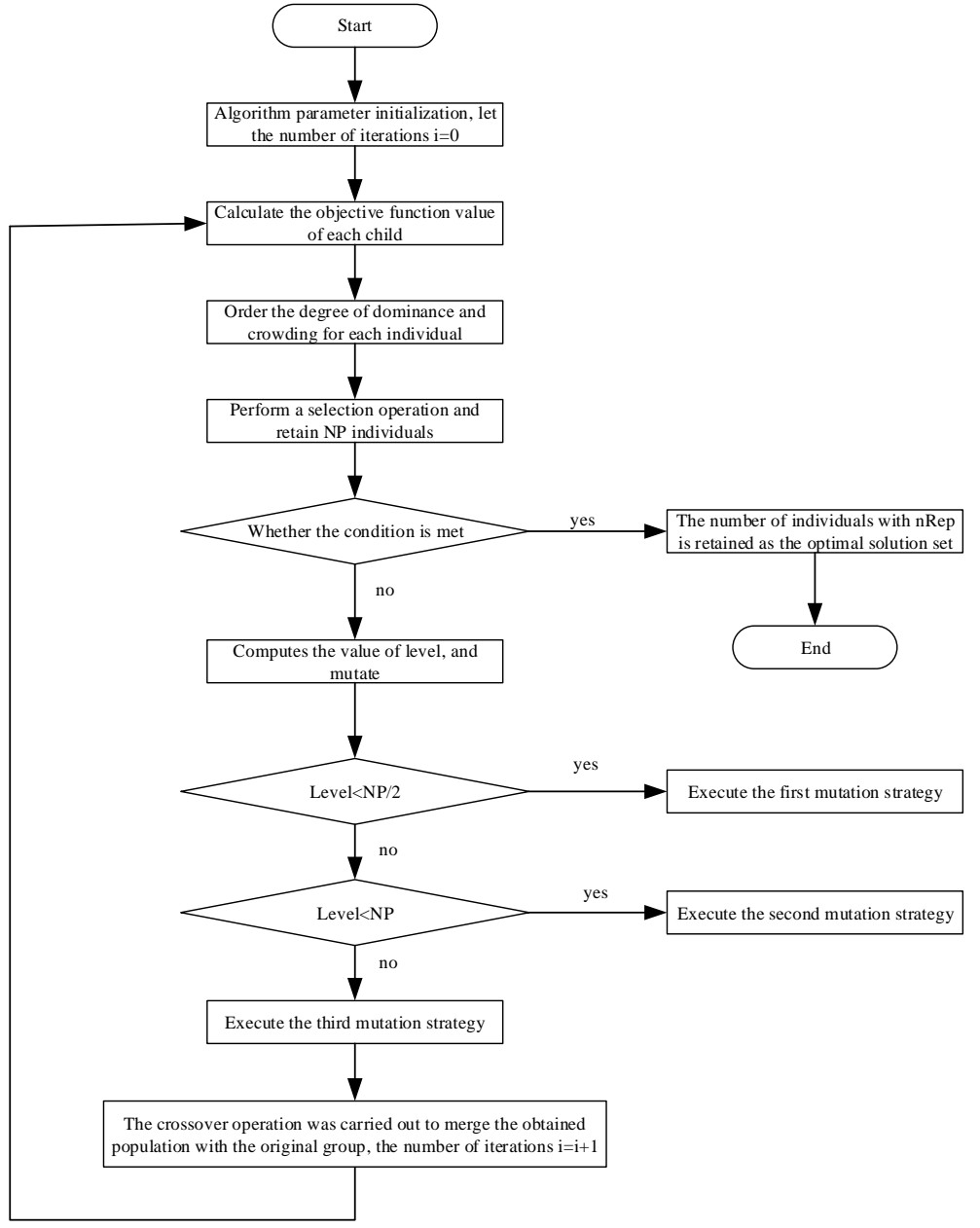

**Figure 3    Flow chart of solving multi-objective optimization problem.**

Where 'n' represents the number of obtained Pareto solutions and $d_i$ signifies the Euclidean distance between the ith solution and the nearest real Pareto optimal solution.

Furthermore, the spatial measure (SP) is the chosen distributivity evaluation index. The formula for SP is depicted as follows:

$$SP = \sqrt{\frac{1}{n-1}\sum_{i=1}^{n}(\overset{\tilde{d}}{-}d_i)^2} \tag{13}$$

Where $n$ is the number of Pareto solutions sought, $d_i$ the minimum Euclidean distance between the ith solution and the other individuals in the current solution set is.

## Parameterization

In this article's $\lambda_P \lambda_U \lambda_Q$ scheme, the penalization factors and their selection are too large to lead to overfitting and too small to converge to the feasible domain.

In this article, we choose $\lambda_P = 100$, $\lambda_U = 300$, $\lambda_Q = 500$. In addition, we choose $F_0$ as 0.25. $CR_{max}$ and $CR_{min}$ are 0.9 and 0.1, respectively, and $F_0 = 0.25$.

The parameters of the algorithm AG-MOPSO are $c_1 = 1, c_2 = 2, w\max = 0.6, wmin = 0.2$, and $nGrid = 8$.

The crossover rate of the algorithm INSGA-II is Pc $= 0.7$, and the variance rate is Pm $= 0.1$. The parameters of the algorithm NSDE are $F_0 = 0.25, CR_{max} = 0.9, CR_{min} = 0.1$. 0.1.

All algorithms retain a consistent number of Pareto optimal solutions set to 30 (step = 30) to ensure fairness in the test results; each algorithm operates with a population size of 45 and undergoes 100 iterations uniformly. For each of the 4 test functions, every algorithm is independently run 20 times. The experimental results are derived from the mean and standard deviation of GD and SP obtained across these independent runs.

## Model comparison

The comparative analysis and evaluation of AG-MOPSO, INSGA-II, and NSDE with the algorithms in this study reflect the performance of convergence evaluation metrics on the test functions ZDT1 and ZDT2, as depicted in Fig. 4. The term "mean" denotes the average value.

It's observed that the distribution of the Pareto solution set generated by the multiobjective particle swarm algorithm, utilizing the adaptive grid method, doesn't match the distributivity achieved by algorithms like NSDE, governed by cyclic congestion degree. Specifically, the mean values of NSDE on ZDT1 and ZDT2 are 0.0018 and 0.0049, respectively. During these instances, the convergence evaluation indexes are recorded at 0.00048 and 0.0038, respectively.

Furthermore, the performance of the constructed and enhanced multiobjective optimization algorithm in this study surpasses NSDE in terms of convergence. Specifically, the GD values of this algorithm are smaller than the previous three algorithms, showcasing better convergence. On ZDT1 and ZDT2, this algorithm's convergence evaluation indexes are 0.000938 and 0.0034, respectively. Remarkably, the average value of ZDT2 reaches 0.0029, notably lower than that of AG-MOPSO, INSGA-II, and NSDE, which are 0.0003, 0.10004, and 0.0014, respectively. The reduction of the mean means that the solutions are more evenly distributed in the target space, which helps to ensure that the convergence distribution in the power financial information management also has better uniformity. At the same time, it also means that the solutions are more dense in the target space, which helps to ensure that the convergence distribution in power financial information

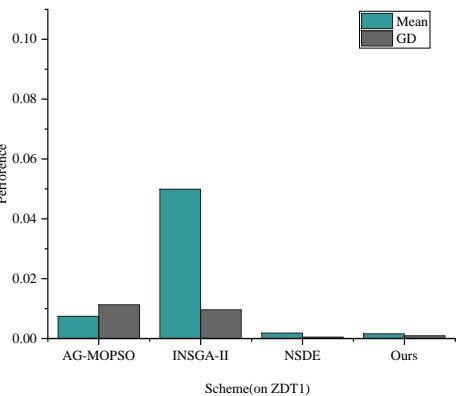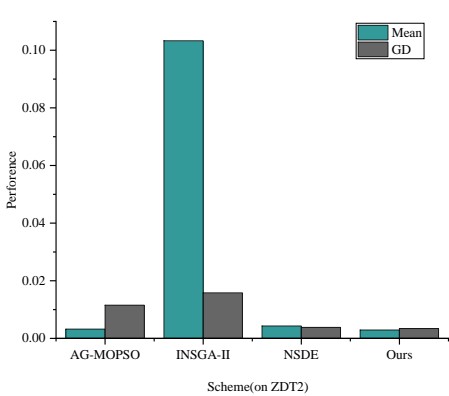

**Figure 4** **The convergence evaluation index performance of each scheme on ZDT1 and ZDT2.**

management has a higher quality. The convergence distribution of the proposed algorithm in power financial information management will be more robust.

The comparison of distributional evaluation metrics based on the performance of convergence evaluation metrics is demonstrated in Fig. 5 through experiments conducted on each of the four test functions.

Observing the average values and the results of distributional evaluation indexes on ZDT1 and ZDT2, it is evident that the SP values of this scheme and NSDE are similar in these functions. Specifically, on ZDT1, the SPs of this scheme and NSDE are 0.0018 and 0.0025, respectively. Notably, the SP of this article's scheme is smaller than that of NSDE, indicating a superior distributability on ZDT1 with a more even distribution.

However, on ZDT2, the SPs of this scheme and NSDE are 0.0026 and 0.0023, respectively. Here, the SP of this article's scheme is more significant than that of NSDE, suggesting a slightly lesser distributional uniformity than NSDE on ZDT2. Further analysis of the results of ZDT3 and ZDT4 shows that the mean values of AG-MOPSO and INSGA-II have changed somewhat, and the mean values of AG-MOPSO on ZDT3 and ZDT4 are 0.315 and 0.1286, respectively. The mean value of INSGA-II on ZDT3 and ZDT4 is 0.6673 and 0.0721, respectively, while the mean value and SP value of the proposed scheme are much smaller than that of the comparison scheme; this indicates superior distributability performance across most test functions, showcasing more even distributions for the proposed model.

The iteration curve of the multiobjective optimization algorithm, showcased in Fig. 6. delineates the incremental enhancement of the solution throughout each iteration. The curve originates from a random solution at the outset of the multiobjective differential evolutionary algorithm. As the algorithm navigates through its search process, the quality of the optimal solution gradually ascends. The solution incrementally approaches or achieves an improved optimal state, eventually reaching a locally optimal solution around the 70th iteration, and this implies that within the current search area, no further superior solution can be found. Ultimately, the algorithm converges to the globally optimal solution.

As iterations progress, the curve gradually converges the solution quality to approximately 0.002, indicating a deceleration in solution enhancement until it reaches a

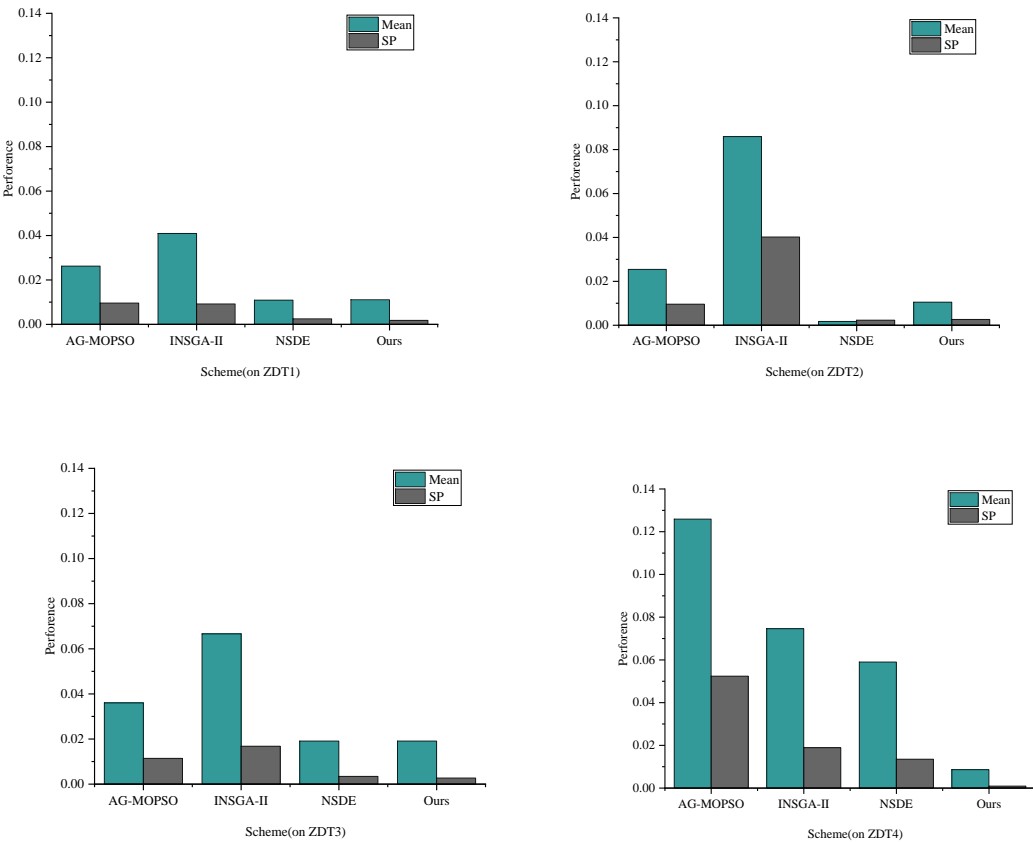

**Figure 5** **The performance of different models under four test functions.**

stable state. The curve depicts that the improvement in the solution gradually slows down, signifying the attainment of a stable state.

The representation in Fig. 7 contrasts the true Pareto of the ZDT1 and ZDT2 test functions, aligned against the Pareto optimal solutions achieved by this article's scheme using the specified parameters. The orange line segments in the visualization depict the optimized Pareto solution set obtained through this article's scheme. At the same time, the red dots represent the true Pareto frontiers derived from the functions' authentic optimal solution set data.

Remarkably, the figure illustrates that the Pareto optimal solution set optimized by the algorithm aligns closely with the actual Pareto front. Moreover, the distribution of the optimal solution set across the Pareto front appears relatively uniform, which validates the algorithm's strong convergence and distribution qualities, affirming the correctness of the algorithm improvement and its efficacy.

## Ablation experiments

In the pursuit of further evaluating the performance of the enhanced multiobjective optimization algorithm, ablation experiments were conducted in this subsection, with

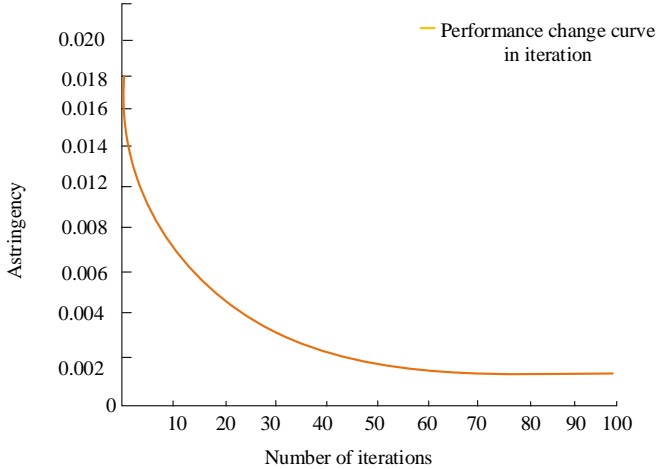

**Figure 6**   **Iterative process.**

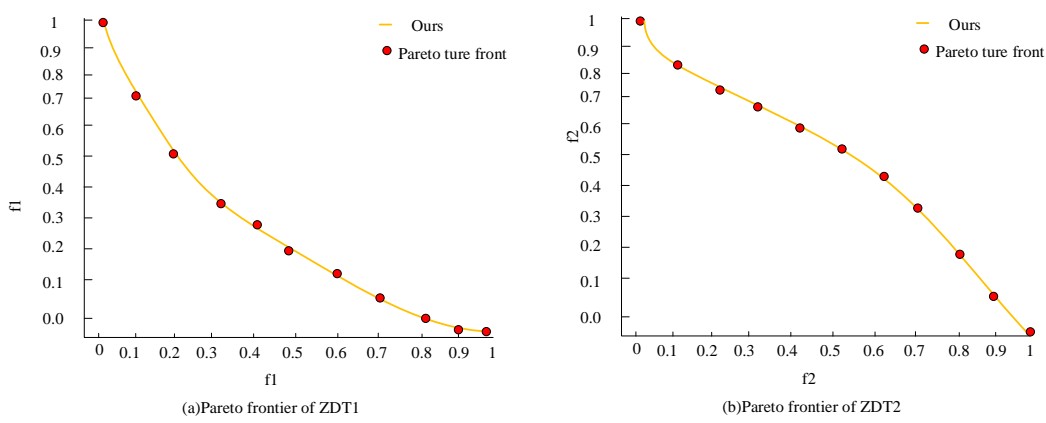

(a)Pareto frontier of ZDT1          (b)Pareto frontier of ZDT2

**Figure 7**   **(A–B) Comparison between the real Pareto frontier and the Pareto optimal solution.**

results displayed in Fig. 8. These experiments aimed to gauge the impact of removing the improved modules within the algorithm.

M1 signifies the outcome when neither equation constraints nor inequality constraints are employed. This absence of constraints leads to challenges in achieving improved results during the genetic algorithm's optimization process, primarily because the state variables in power system optimization remain unconstrained.

M2 and M3 represent outcomes when adaptive variability and operator dynamic cross factors are excluded or included independently. Under M2, GD and SP measure at 0.0091 and 0.0063, respectively. For M3, GD and SP values are recorded at 0.0042 and 0.0045, respectively. These results underscore the significance of adaptive mutation in enhancing the performance of the multiobjective optimization algorithm. Adaptive mutation plays

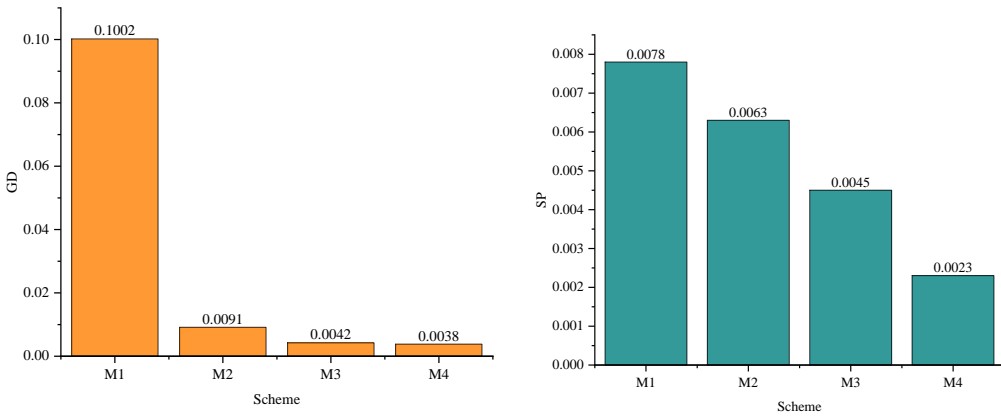

**Figure 8** Results of ablation experiment.

a critical role in driving performance improvements within the algorithm. M4 represents the proposed model.

The simultaneous integration of economic, environmental, and social objectives within the power system is a complex and multifaceted task that requires the integration of multiple factors. We employ an integrated multiobjective optimization approach that combines advanced technical means and decision-support tools to achieve this.

First, we build the information management model. We use the fuel cost of the thermal power generator set, active power network loss of power system, and voltage quality as the optimization objectives of the intelligent power financial system and establish the mathematical model of each objective. In this model, we can use various indicators for cost-benefit analysis, environmental and social impact assessments, *etc.* At the same time, taking into account multiple constraints in the power system, such as fuel cost, power system active network loss, voltage quality, *etc.*, the adjustment of these parameters is integrated with the goal of economical and efficient development, protection of the energy environment and socially sustainable development. Using a multiobjective optimization algorithm with adaptive variation, we can find solutions that maximize economic, environmental, and social benefits under all constraints.

## DISCUSSION

The analysis outlined in 'Model comparison' and 'Ablation experiments' demonstrates the enhanced efficacy of our refined multiobjective optimization algorithm across all four ZDT1, ZDT2, ZDT3, and ZDT4 test functions. Simultaneously, the augmentation of the Pareto optimal solution's performance in our multiobjective optimization algorithm has been a consistent endeavor, and this has been achieved by integrating non-equation and inequality constraints governing the state variables within power system optimization. Additionally, adaptive variational and operator dynamic crossover factors have been instrumental in this optimization. In electric power financial systems, the optimization of information management is essential to intelligent processing, and its ultimate goal is

to improve the allocation efficiency of material and financial resources. The improved multiobjective optimization algorithm can adequately deal with multiple conflicting objectives and show adaptability in uncertain factors and abnormal situations. This ability enables decision-makers to consider issues more comprehensively and carefully and make more informed decisions, significantly improving decision-making efficiency.

With the continuous development of the power market and the increasingly fierce competition, the requirements for financial management of electric power enterprises are also increasing. The information management model based on the multiobjective optimization algorithm we built provides more comprehensive and accurate financial analysis and decision support for electric power enterprises. Through the extensive study of historical and real-time data, the model can provide more accurate financial risk assessment and early warning for electric power enterprises, thus significantly enhancing financial risk management ability.

This enhancement of risk management ability helps power enterprises achieve sustainable development, promotes technological innovation, and upgrades the entire power industry. By optimizing the management of power financial information, we can improve the rationality of resource allocation and the competitiveness and development level of the power industry. In this study, we design the objective function based on three key objectives: fuel cost, active network loss, and voltage quality of the power system. This design method not only helps us understand the complexity of power financial information management more profoundly but also can optimize the allocation of resources, such as material, financial, and other vital resources. In this way, we support the intelligent development of power financial information management, laying a solid foundation for the future development of the power industry.

Considering that only three algorithms were chosen for improvement and problem-solving, as the number of objectives expands, it becomes imperative to delve into novel enhancements for intelligent multiobjective optimization algorithms. This exploration aims to guarantee robust convergence and distribution in these algorithms. At the same time, power financial information management is a continuous process that must be dynamically adjusted and optimized according to the actual situation. Therefore, we will discuss the effect of other intelligent algorithms to solve the multiobjective problem of the power financial system in the future and update and optimize management strategies in real-time according to the changes in the power market, policy adjustments, and other factors to ensure the accuracy and effectiveness of the power financial information.

## CONCLUSION

This study focuses on enhancing the multiobjective differential evolutionary algorithm concerning optimizing fuel cost, active network loss, and voltage quality within the power financial system. To address this, mathematical models corresponding to each objective have been formulated. Equation and inequality constraints are introduced, with state variables constrained through penalty functions. The differential evolutionary algorithm is then applied for refinement, optimizing the algorithm's parameters considerably.

Subsequently, experiments utilizing standard test functions are conducted, showcasing the algorithm's enhanced performance resulting from these refinements. The Pareto frontiers of the power system's two-dimensional and three-dimensional objective spaces are computed for each algorithm. Comparative analysis between the improved differential evolution algorithm and its predecessor demonstrates the superior efficacy of the refined approach. The experiments validate each algorithm's effectiveness in resolving the multiobjective optimization challenges within the power financial system.

### Funding
The authors received no funding for this work.

### Competing Interests
Junhui Hu is employed by the State Grid Ningbo Electric Power Supply Company. Hongxiang Cai, Shiyong Zhang, Chuanxun Pei and Zihao Wang are employed by the State Grid Ninghai Power Supply Company. The employer has no issues to publish this article.

### Author Contributions
- Junhui Hu conceived and designed the experiments, analyzed the data, authored or reviewed drafts of the article, and approved the final draft.
- Hongxiang Cai conceived and designed the experiments, performed the computation work, prepared figures and/or tables, and approved the final draft.
- Shiyong Zhang conceived and designed the experiments, analyzed the data, prepared figures and/or tables, and approved the final draft.
- Chuanxun Pei performed the experiments, performed the computation work, authored or reviewed drafts of the article, and approved the final draft.
- Zihao Wang performed the experiments, analyzed the data, authored or reviewed drafts of the article, and approved the final draft.

### Data Availability
The data is available in at Zenodo: Tushar Chugh. (2019). Electric Power Assisted Steering ESR 2 (Final) [Data set]. Zenodo. Available at https://doi.org/10.5281/zenodo.3263796.

The code is available in the Supplementary Files.

### Supplemental Information
Supplemental information for this article can be found online at http://dx.doi.org/10.7717/peerj-cs.2023#supplemental-information.

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
