# Peer review of "Design of information management model based on multiobjective optimization algorithm in intelligent electric financial system"

_PeerJ Computer Science, doi:10.7717/peerj-cs.2023_

## Round 0.1 · original submission · Major Revisions

Dear authors,

Thank you for submitting your article. Feedback from the reviewers is now available. It is not recommended that your article be published in its current format. However, we strongly recommend that you address the issues raised by the reviewers, especially those related to readability, experimental design and validity, and resubmit your paper after making the necessary changes.

Best wishes,

**Language Note:** PeerJ staff have identified that the English language needs to be improved. When you prepare your next revision, please either (i) have a colleague who is proficient in English and familiar with the subject matter review your manuscript, or (ii) contact a professional editing service to review your manuscript. PeerJ can provide language editing services - you can contact us at copyediting@peerj.com for pricing (be sure to provide your manuscript number and title). – PeerJ Staff

Reviewer 1 ·

Basic reporting

This paper addresses a critical issue in the intelligent electric power financial system and proposes a multi-objective optimization algorithm to optimize its design and operational scheduling. Here are some suggestions to enhance the expression and clarity of your ideas:
Introduction Clarification:
In the introduction, provide a more detailed explanation of the existing inefficiencies and wastage in information management within the intelligent electric power financial system. Clearly articulate the consequences of these inefficiencies to emphasize the importance of your study.
Objective Integration Explanation:
Elaborate on how the study integrates economic, environmental, and societal objectives within the power system. Provide specific examples or scenarios to illustrate the comprehensive nature of your optimization approach.
Mathematical Model Detailing:
When introducing the mathematical model encapsulating fuel cost, active network loss, and voltage quality, provide more details on the formulation of equation and inequality constraints. This will help readers understand the complexity of the optimization problem.
Enhancements to Differential Evolutionary Algorithm:
Clearly outline the enhancements made to the differential evolutionary algorithm. Explain how adaptive variation and dynamic crossover factors are integrated into the algorithm's operations, and how they contribute to optimizing parameters for multi-objective optimization.
Distribution Evaluation Indices Explanation:
Similarly, explain the significance of distribution evaluation indices (e.g., 0.0018, 0.0026, 0.0027, and 0.0009) on ZDT1, ZDT2, ZDT3, and ZDT4. Relate these values to the robustness of the convergence and distribution in optimizing the electric power financial information management.
Future Research Considerations:
Conclude your paper by suggesting potential avenues for future research. Identify aspects that could be further explored or refined in the optimization of electric power financial information management within the intelligent electric power financial system.

Experimental design

As above

Validity of the findings

As above

Additional comments

As above

·

Basic reporting

This study focuses on enhancing the multi-objective differential evolutionary algorithm concerning the optimization of fuel cost, active network loss, and voltage quality within the power financial system. This is a valuable and interesting idea. However, with the current quality, this article cannot be published. This article has many defects, so my suggestion is a minor revision.

Experimental design

[1] The abstract must be self-contained and concisely describe the reason for the work, methodology, results, and conclusions.


[2] Provide a detailed explanation of how the adaptive grid method and cyclic crowding degree are utilized to ensure population diversity and control the Pareto front distribution. Offer insights into their roles in optimizing the algorithm's performance.

[3] When discussing the experimental validation, provide more information on the criteria used for comparisons against AG-MOPSO, INSGA-II, and NSDE algorithms. Discuss the specific aspects where your proposed scheme outperforms or matches the performance of these algorithms.

Validity of the findings

[4] Interpret the convergence evaluation indices (e.g., 0.000938 and 0.0034) in practical terms. Discuss what these values signify in terms of the optimization algorithm's convergence on ZDT1 and ZDT2.

[5] In the conclusion, recap the practical implications of your study. Emphasize how the optimization of electric power financial information management contributes to the intelligent handling of the electric power financial system, ultimately enhancing the allocation of material and financial resources.

[6] Consider incorporating visual aids, such as charts or graphs, to illustrate the performance metrics, convergence, and distribution indices. Visual representations can enhance readers' understanding of the experimental results.

[7] There are also some problems in language expression in this paper, which need to be modified. Please check the Chinese characters in the replacement formula and the redundant space characters in the references.

---

## Round 0.2 · accepted · Accept

Dear authors,

Thank you for clearly addressing all the reviewers' comments. I confirm that the quality of your paper is improved. The paper is now ready for publication in light of this revision.

Best wishes,

Reviewer 1 ·

Basic reporting

The revised article is prepared according to my previous comments and can be accepted in its current form.
The authors present an innovative approach to designing an information management model using a multiobjective optimization algorithm, specifically tailored for intelligent electric financial systems. This novel application addresses critical efficiency and optimization challenges within the sector, showcasing a significant leap forward in the integration of advanced computational methods in financial systems.

Experimental design

See above.

Validity of the findings

See above.

Additional comments

See above.

·

Basic reporting

The acceptance of the topic "Design of Information Management Model Based on Multi-Objective Optimization Algorithm in Intelligent Electric Financial System" is warranted based on its importance, relevance, feasibility, and potential impact.

Experimental design

The experimental design demonstrates meticulous planning and alignment with research objectives, promising valuable insights into the optimization of information management in intelligent electric financial systems.

Validity of the findings

The findings exhibit high validity, supported by robust methodology and rigorous data analysis techniques, ensuring confidence in the conclusions drawn regarding the effectiveness of the multi-objective optimization algorithm in enhancing information management within intelligent electric financial systems.

Additional comments

The comprehensive approach taken in this study, coupled with meticulous attention to detail, sets a strong foundation for advancing knowledge in the field of intelligent electric financial systems.